# Die-Level Thinning for Flip-Chip Integration on Flexible Substrates

**Muhammad Hassan Malik [1,2], Andreas Tsiamis [3], Hubert Zangl [2,4], Alfred Binder [1], Srinjoy Mitra [3] and Ali Roshanghias [1,*]**

1   Silicon Austria Labs GmbH, Europastrasse 12, A-9524 Villach, Austria; muhammad-hassan.malik@silicon-austria.com (M.H.M.); alfred.binder@silicon-austria.com (A.B.)
2   Institute for Smart Systems Technologies, Alpen-Adria-Universität Klagenfurt, A-9020 Klagenfurt, Austria; hubert.zangl@aau.at
3   School of Engineering, Institute for Integrated Micro and Nano Systems, University of Edinburgh, Edinburgh EH9 3FF, UK; a.tsiamis@ed.ac.uk (A.T.); srinjoy.mitra@ed.ac.uk (S.M.)
4   Alpen-Adria-Universität Klagenfurt-Silicon Austria Labs, Ubiquitous Sensing Systems Lab, A-9020 Klagenfurt, Austria
*   Correspondence: ali.roshanghias@silicon-austria.com

**Abstract:** Die-level thinning, handling, and integration of singulated dies from multi-project wafers (MPW) are often used in research, early-stage development, and prototyping of flexible devices. There is a high demand for thin silicon devices for several applications, such as flexible electronics. To address this demand, we study a novel post-processing method on two silicon devices, an electrochemical impedance sensor, and Complementary Metal Oxide Semiconductor (CMOS) die. Both are drawn from an MPW batch, thinned at die-level after dicing and singulation down to 60 µm. The thinned dies were flip-chip bonded to flexible substrates and hermetically sealed by two techniques: thermosonic bonding of Au stud bumps and anisotropic conductive paste (ACP) bonding. The performance of the thinned dies was assessed via functional tests and compared to the original dies. Furthermore, the long-term reliability of the flip-chip bonded thinned sensors was demonstrated to be higher than the conventional wire-bonded sensors.

**Keywords:** ultra-thin-chips; multi-project wafers; hybrid integration; thermoconic flip chip; anisotropic conductive adhesives; flexible electronics; flip chip bonding

## 1. Introduction

### 1.1. Flexible CMOS Sensors with Die-Level Post-Processing

Flexible and conformal systems are one of the major research areas in today's electronics industry. If a sensor and its associated silicon-based readout electronics become flexible, the utility of the device goes far beyond the traditional applications based on rigid printed circuit boards [1–3]. This includes biomedical, environmental, and industrial applications. One way of doing this is to develop novel flexible materials and substrates for both transducers and electronics [4]. Though there has been quite some progress in this field [5], they are far from the abilities of modern analog and digital electronics in traditional CMOS substrates. Hence, flexible systems that can use CMOS chips provides a promising possibility. Furthermore, if the transducer can be integrated in the close vicinity of a CMOS chip (e.g., system in package (SiP) configuration), the device becomes ideal for several applications, such as patch sensors and large-area electronics. However, if it is an electrochemical sensor system exposed to a harsh environment (e.g., medical implants, water/fluid sensors), hybrid bonding, encapsulation, and packaging become critical. A reliable process that can achieve a decent yield for early stage prototyping is essential, and standard wirebonding techniques are not suitable for this purpose.

In our previous study, flip-chip integration of ultra-thinned chips (UTC) down to 10 µm in flexible printed electronics was successfully demonstrated [6]. It was shown that

in flexible hybrid electronics, the thinner the die, the higher the reliability. This is because a thinner die provides superior flexibility and can withstand higher bending stresses [6–8]. In order to achieve the system illustrated in Figure 1, in this paper, we demonstrate how ultra-thin silicon sensors and an ultra-thin CMOS chips can be flip-chip bonded on a flexible substrate. The ultra-thin sensors are also subjected to a harsh environment, and various bonding techniques are compared for reliability. In [4], the thinning and dicing of the test wafers with simple daisy chain structures were performed in a standard wafer-level process by using the Stealth Dicing Before Grinding (SDGB) process. The average contact resistance and possible short/open circuits were the only measures for those test chips, whereas more advanced functional tests were not possible. However, advanced functional components, such as Complementary Metal Oxide Semiconductor (CMOS) devices, image sensors, and silicon interposers, are either not an affordable option for R&D institutes or only offered by foundries in certain wafer diameters (300 mm), which can not be post-processed by R&D facilities. As a result, silicon foundries offer an attractive and low-cost option through the multi-project wafer (MPW) service, where instead of wafers, customers receive single chips with their IC design [9–11]. Although MPW decreases the cost of fabrication, in turn it makes the individualized post-processing potentially more complicated, as most process tools are designed for wafer-level processes. Nevertheless, there are also many technology options and processing methodologies to facilitate individualized post-processing. For instance, we have recently reported a spray-based photoresist deposition combined with optical maskless lithography for patterning MPW chips as opposed to conventional spin-coating and using photomasks [5]. Thinning of single chips is also challenging and requires modifications to standard wafer thinning. However, in an MPW, some devices require thinning while others do not [12]. The wafer is first diced and then the individual dies are thinned. Correspondingly, in this study, a die-level thinning route for MPW chips was sought as the only viable option to thin CMOS and sensor chips.

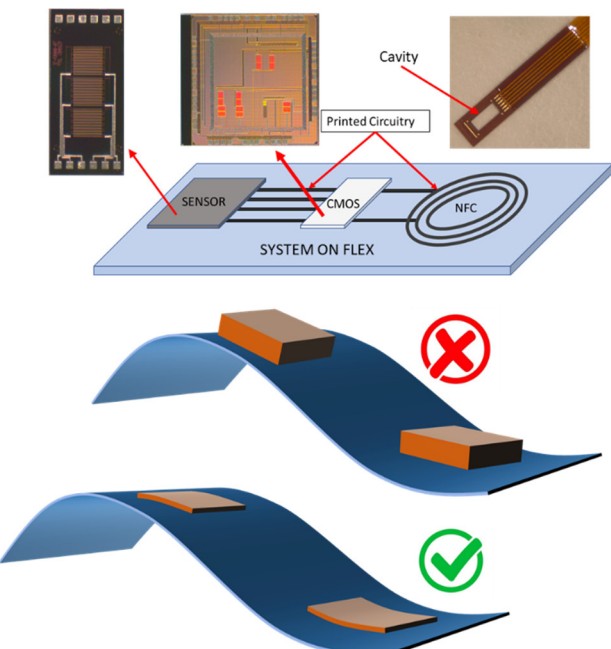

**Figure 1.** Hybrid integration of silicon-based components on flexible substrates; the advantage of ultra-thinned chips to leverage the flexibility of the whole system.

### 1.2. Flip chip Integration of the Thinned Chips

For the integration of the chips on flexible substrates, the conventional lateral chip-on-board (CoB) configuration is being substituted by flip-chip bonding and vertical 3D integration. Flip chip bonding was introduced in the 1960s by IBM, which was the so-called C4 (Controlled Collapse Chip Connect), which consisted of solder bumps on the

chip, which were used for electrical connections instead of the traditional wire bond [13]. At that time, the flip chip was meant to replace wire bonding, which was considered unreliable, less productive, and expensive. Nevertheless, wire bonding was improved over the years, resulting in a more flexible and high-yield process [14,15]. Today, flip chip is applied where semiconductor manufacturers or packaging companies profit from its advantages in size, cost, reliability, and performance over wire bonding [16]. Another major advantage of this technique is that it provides the shortest conductor path essential for the high-frequency application. Flip-chip bonding can be performed by using only temperature (for solder bumps), thermo-compression (mainly assisted by conductive adhesives), or by using thermosonic energy (for Au and Cu bumps) [17]. The thermosonic flip-chip process has the advantage of low bonding force, low temperature, and short bonding time yielding a strong metallurgical joint [18,19]. Furthermore, this process is a clean, lead-free, and dry process which is well suited for hybrid integration of electronic packages [20]. The process is then further aided with an underfill which provides increased mechanical stability to the metallic bond. In the past, this technology has been used in the assembly of acoustic wafer filter packages, MEMS [21,22] and vertical-cavity surface-emitting lasers (VCSELs) [23]. However, the process can still be detrimental to fragile UTCs and also requires manual underfills that can be time consuming. Furthermore, the need for Au/Cu bumps limits whether the UTCs can be processed at die-level.

Thermocompression bonding assisted by anisotropic conductive paste (ACP) is a recently developed and promising flip-chip bonding methodology that can be applied to different pad metallizations with and without stud bumps and has been implemented successfully on flex to PCB bonding, Chip on Glass (COG), Chip on Film (COF) Chip in Film (CIF), and LCDs [6,24–26]. The ACP bonding process consists of trapping the conductive particle between the chip bump and the substrate metallization, resulting in an electrical connection. Heating the die and the substrate is also essential at first to initialize the flow of the ACP and then subsequently curing the adhesive. Furthermore, the free space between the electrodes is occupied by the resin which acts as an insulator (or underfill) that enhances the mechanical and thermal stability of the bonding [25,27]. The current work aims to evaluate both thermosonic and ACP flip-chip techniques for hybrid integration of die-level thinned sensors and CMOS chips. In Figure 1, the hybrid integration of chips on the flexible substrate and the advantage of ultra-thinned chips to leverage the flexibility of the whole system is schematically presented.

## 2. Materials and Methods

### 2.1. Die-Level Thinning

The ~3 mm × 3 mm CMOS chips with an initial thickness of 790 μm were fabricated in a commercial foundry and diced from a 12-inch MPW. The sensor chips (~1.4 mm × 3.0 mm), with an initial thickness of ~500 μm, were fabricated and diced from a 4-inch wafer using standard photolithographic and processing techniques and consist of 3 sets of 500 × 500 platinum interdigitated electrodes (IDE). These were intended to be flip-chip bonded to a flexible substrate (flex-PCB), while the interdigitated area would be exposed inside the cavity for sensing. A flex-PCB with an open cavity to place the sensor chip was designed and fabricated as shown in Figure 1. The IDEs and the flex were designed to provide the easiest electrochemical sensing possible (impedance of an analyte) yet pass it through the rigorous reliability tests and consists of simple functional blocks to verify the viability of using similar post-processing as the silicon IDE-sensors.

Bumping-Thinning Sequence

In order to facilitate flip-chip bonding, bumping of the chip pads is usually done either via under-bump-metalization (UBM) followed by solder bumping or via stud bumping. The latter is more convenient for die-level processing, since it only requires a ball bonder. The Au stud bumping is vital for thermosonic flip-chip bonding; whereas for ACP flip-chip bonding, stud bumping is not necessary but rather advantageous and can mitigate

reliability issues. In this study, both bumping-before-thinning and bumping-after-thinning sequences were examined. The IDE-sensors were first Au stud-bumped by using a manual ball bonder and then coined. Coining guarantees co-planarity, which results in a good flip-chip interconnect on a larger area [28]. The coined sensor chips were then thinned down to $67 \pm 2$ μm. On the other hand, the CMOS chips were first thinned down to $80 \pm 2$ μm. Some of the thinned CMOS chips were bonded without bumping, while some thinned chips were bumped after thinning. The first assessment results showed that both fabrication sequences can be applied for the above-mentioned chips, however it was revealed that stud bumping-after-thinning of the ultra-thinned chips (e.g., less than 60 μm) could generate micro-cracks in the chips due to the induced mechanical stresses upon bumping.

An illustration of the die-level thinning process is presented in Figure 2. Here, a silicon carrier wafer with a UV-curable adhesive film was utilized. The singulated dies (IDE-sensors and CMOS chips) were populated on the carrier wafer, so that a uniform distribution of the chips on the carrier wafer was obtained. Here, auxiliary dummy chips were employed for better population and enhancing the stability of the reconstituted wafer. The reconstituted wafer was consequently ground and polished using a semi-automatic grinder (DAG 810, Disco Hi-Tech Corp., Munich, Germany). During the back-side thinning process, the front-side of the chips was fully covered by the UV-curable adhesive film to avoid damaging to the active surfaces. Finally, the thinned dies were then released from the carrier via UV exposure.

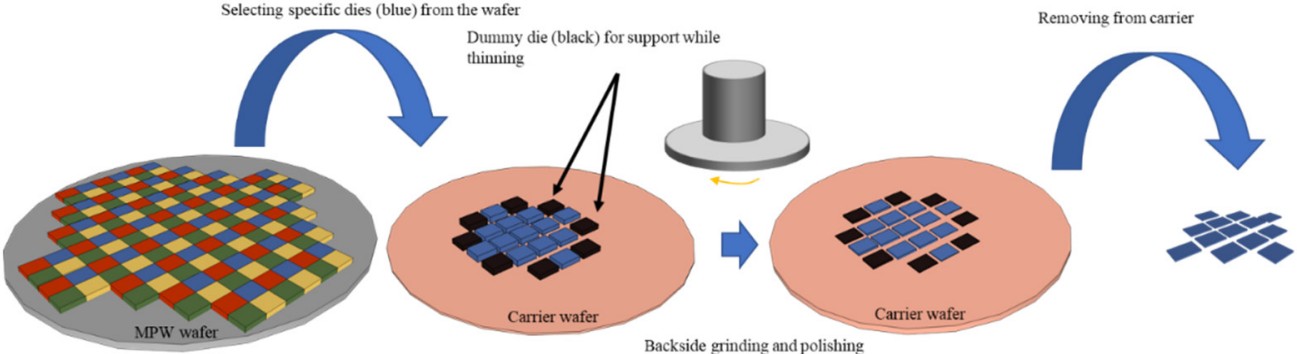

**Figure 2.** Die-level thinning of the multi-project-wafers.

### 2.2. Chip on Board Integration via Wire Bonding

Conventionally, the dies would have been integrated via CoB method by means of wire bonding. In this study, the aim was to compare these standard samples to the flip-chip bonded ones. Process flows to fabricate the conventional packages are complicated, including photolithography and washing of the non-cured epoxy with Acetone and Deionized (Di) water. This process flow is also presented and compared with the flip-chip procedures in Figure 3. For wire bonding, the die was first attached to the flex with an adhesive, followed by connecting the pads on the die to the pads on the flex with wire bonding. The whole structure, including the bonding wires, is then encapsulated with a UV-cured epoxy (glob top). Before curing the epoxy, a mask is placed in front of the sensor area to avoid curing the epoxy in the specific region. The unmasked area was UV-cured, and uncured epoxy was washed away to expose the sensor area on the die. As a result, the cavity created with this method was 1.85 mm (Length) × 0.8 mm (Width) × 0.38 mm (Height), which was comparable to the cavity of the flex-PCB for flip-chip bonding: 1.95 mm (Length) × 0.8 mm (Width) × 0.325 mm (Height).

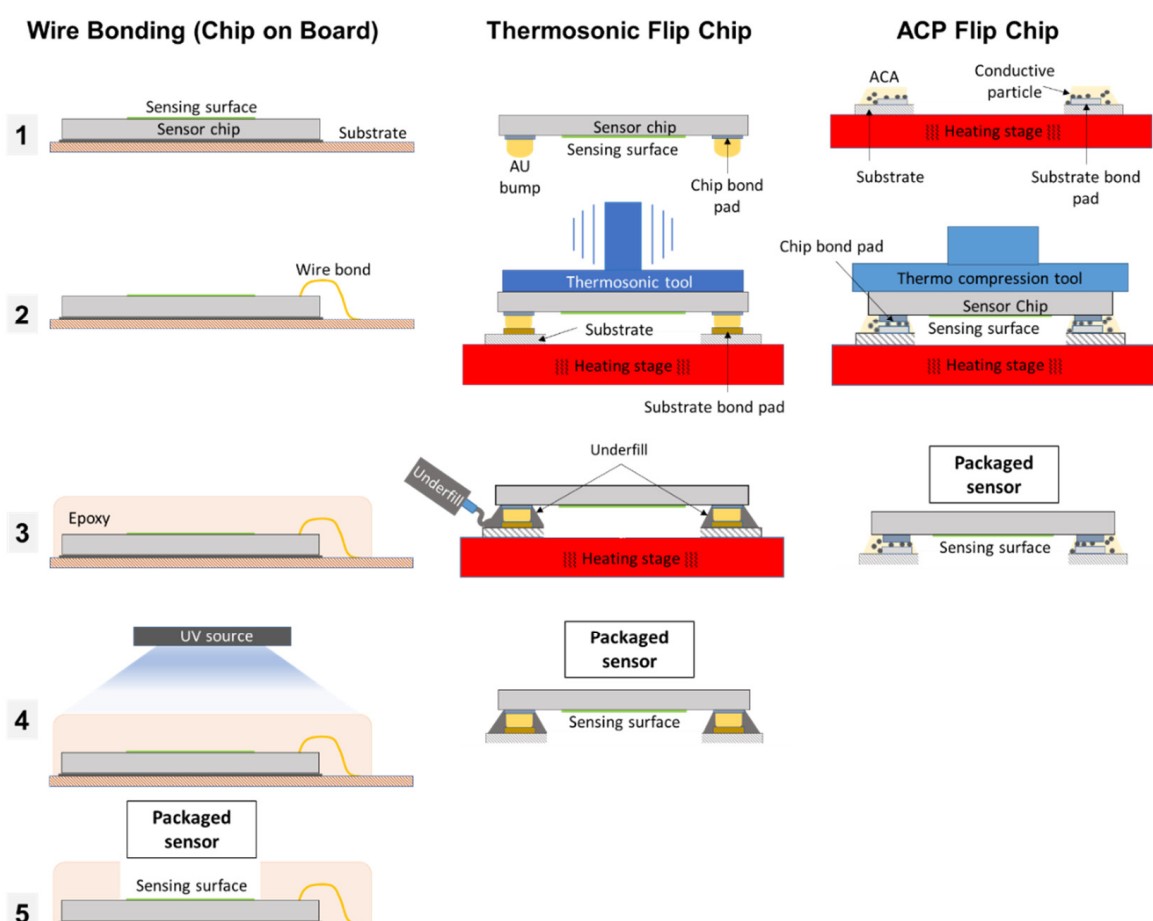

**Figure 3.** Schematic comparison among the ACP, thermosonic flip chip, and the wire bonding integration process flow to realize a sensor package with an exposed cavity. The sensing area of the chip is colored green.

### 2.3. Flip-Chip Integration

For flip-chip integration of the thinned dies on flex, two methodologies were exerted (ACP and thermosonic bonding), as shown in Figure 3. The bonding process was carried out employing a micro-assembly station (Fineplacer®, Finetech GmBH, Berlin, Germany). The micro-assembly station was equipped with both thermo-compression and thermosonic bonding heads as well as a micro-dispensing unit.

#### 2.3.1. Thermosonic Bonding

The ultrasonic module had an output power of 20 W, a driving frequency of 60 kHz, and a maximum bonding force of 20 N. For thermosonic bonding, the dies with Au stud bumps were used, whereas for ACP bonding, both bumped and un-bumped chips were employed. The total process time was 10 s with ultrasonic vibration applied for 500 ms. The force used was 5 N with ultrasonic power at 2 W. The temperature of the substrate was kept at 220 °C. At higher static forces and ultrasonic powers, microcracks in the ultra-thin sensor chips were observed. As inferred from Figure 3, thermosonic bonding was a three-step process starting with bumping, followed by bonding. After the bonding, an underfill material was dispensed and thermally cured to provide mechanical stability to the bonds.

#### 2.3.2. ACP Bonding

To carry out a successful flip-chip bonding, a dedicated procedure for die picking from the transfer film, flipping, alignment, placement, and bonding was developed. The parameters for a successful bonding were deduced with a combination of a variety of

temperatures of the tool and the stage coupled with an optimum force that can realize a successful bond without damaging the thin die [6,25]. The bonding parameters were optimized so the stage temperature and the tool temperature were 170 °C. A low bonding force (60 N) was selected to avoid possible damages to the thinned dies. Planarization of the surfaces is critical in this process, therefore the picking tool and the stage were precisely adjusted with the help of a pressure-sensitive foil, which results in a complete planer chip orientation. This process is a two-step process, starting with dispensing the ACP followed by applying pressure and temperature. This is illustrated in Figure 3. Here, a commercial anisotropic conductive paste (ACP, DELO® MONOPOX AC268, Windach, Germany) was used. The ACP consists of an epoxy resin glass transition temperature ($T\_g$) of 153 °C and conductive nickel particles with an average diameter of 5 μm. Here, the bonding was performed with both unbumped and bumped dies.

## 3. Results and Discussion

### 3.1. Silicon IDE-Sensors

Figure 4a,b shows SEM images of both thick and thinned sensor chips with the final thickness of 67 μm. The initial thickness of the chips was ~500 μm. The thinning was conducted uniformly and the total thickness variation (TTV) among the chips was less than 2 μm. The coined Au stud-bumps that are used for flip-chip bonding are illustrated in Figure 4c. Finally, the cross-sectional SEM images of the flip-chip bonded sensor chips via thermosonic (e) and ACP bonding (d) are presented in this figure. As seen in the case of thermosonic bonding, direct bonding between the Au bump and the electroless nickel immersion gold (ENIG) pad of the flex was generated, whereas, in the ACP, the nickel particles in the paste facilitate the electrical path. Figure 5 shows the placement of the IDE-sensor on the flex, where it was flip-chip bonded on the periphery of the cavity and the sensing area was exposed to interact with different analytes. As mentioned, the flex strip had a cavity at the tip. In order to encapsulate the pads and the bumps, underfill was used after thermosonic bonding. In the case of ACP bonding, the ACP itself could provide encapsulation, rendering a straightforward and one-step bonding and encapsulation procedure.

### 3.2. Impedance Analysis

Both thermosonic-bonded and ACP-bonded sensor samples were first assessed in air and then immersed in water and phosphate-buffered saline (PBS) solution. The measurement results are plotted in Figure 6. The plots show typical characteristics of the measured impedance across a frequency range for the media used and the IDE sensor material and dimensions. It can be inferred that the impedance of the bonded samples has negligible deviation between the two bonding techniques. This trend is visible in the full frequency range from 100 Hz to 1 MHz. All the samples (three samples per kind) were functional without any faulty bonding.

### 3.3. Comparison to the Wire-Bonded Sensors

It can be deduced from Figure 3 that CoB sensor packages require more steps compared to flip-chip packages. Also, it can be concluded that ACP bonding is the process with the minimum number of process steps. Figure 7 shows the measured impedance of samples submerged in PBS that have been bonded using the three different techniques: wire bond, thermosonic flip-chip, and ACP. As inferred from this figure, all three methods exhibited similar impedance behaviour with slight deviation. It is noteworthy to metion that the generated caivites in the above-mentioned methods could not ensure a same exposure area of the IDE electrode. This is likely to be one cause for the slight impedance variations. For instance, it can be seen in Figure 5d that the manually dispensed adhesive might cover some parts of the electrodes. This issue can be solved via further optimization of the dispensing process by using an automated dispensing robot making a cavity that is much larger then the electrodes.

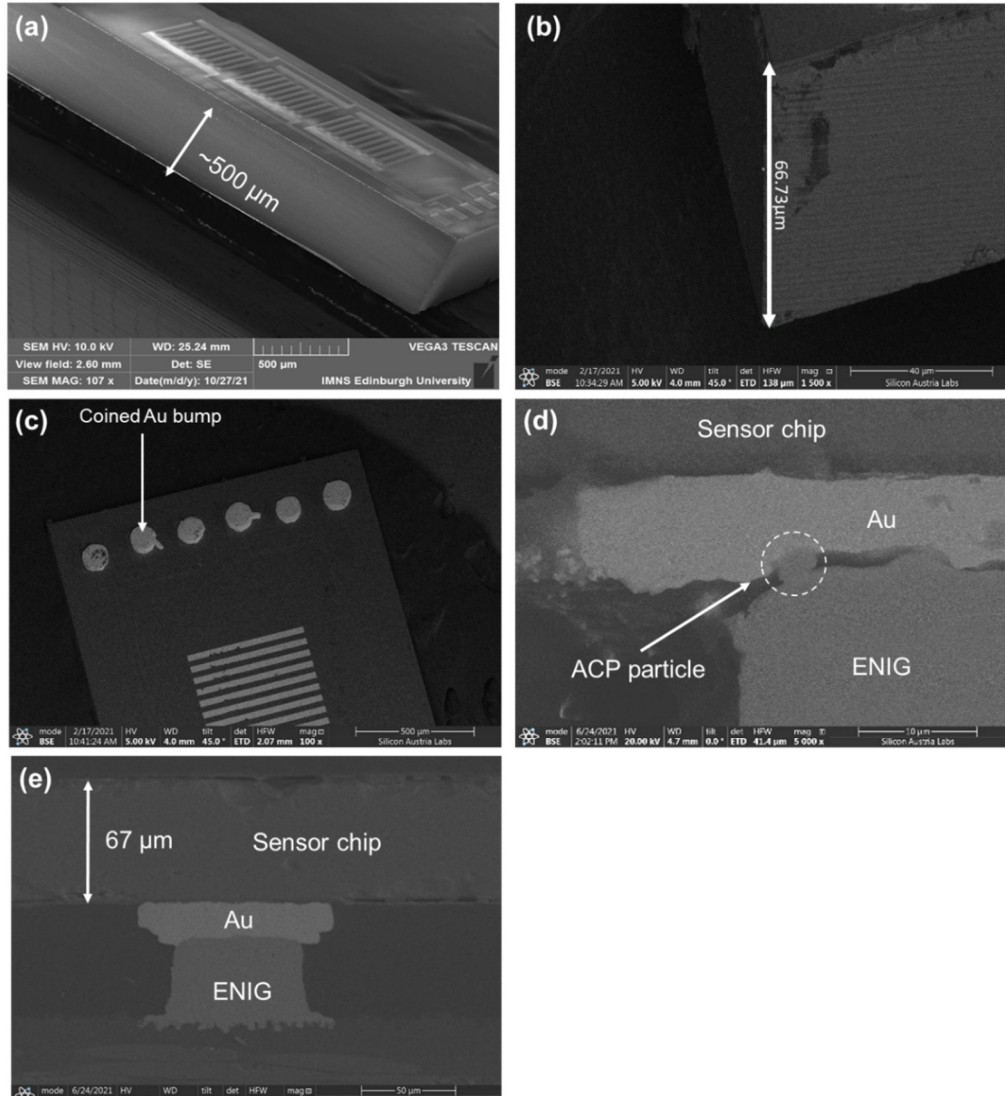

**Figure 4.** SEM images of the thinning and flip-chip integration of the sensor chip on flex, (**a**) thick sensor chip, (**b**) thinned sensor chip, (**c**) cocned Au bump on chip, (**d**) ACP flip-chip bonding, and (**e**) thermosonic flip-chip bonding on flex.

To determine long-term reliability, all the samples were characterized for an extended period spanning several days. The impedance was measured everyday for all three types of samples. The impedance variation of thinned IDE sensors for a span of days at 1 MHz is illustrated in Figure 8. Here, the average impedance variation is shown with a line, whereas the failed samples are pinpointed with "x". As seen, the wire-bonded samples failed much earlier within the first 10 days. Concerning flip-chip bonded samples, it was observed that the impedance of both samples slightly varied with time, however, they were all still functional up to one month. Throughout the period of testing, both samples exhibited the same behavior, and the impedance change was similar. After a mean time of 30 days, the sensor's impedance increased spontaneously, as implied by Figure 8. The samples were then taken out of the PBS and investigated under an optical microscope. The failure could be attributed to either degradation of the sensors or the package and is still under investigation. However, a 30-day mean lifetime is good enough for multiple acute medical implants (e.g., anatomic leaks, wound healing) where the recovery of an organ is monitored for 2 to 3 weeks immediately after surgery. Similarly, monitoring of hazardous and hard-to-reach environments (e.g., in processing plants) require sensors mounted on flexible robots that need to be in place for a few days before problems can be fixed.

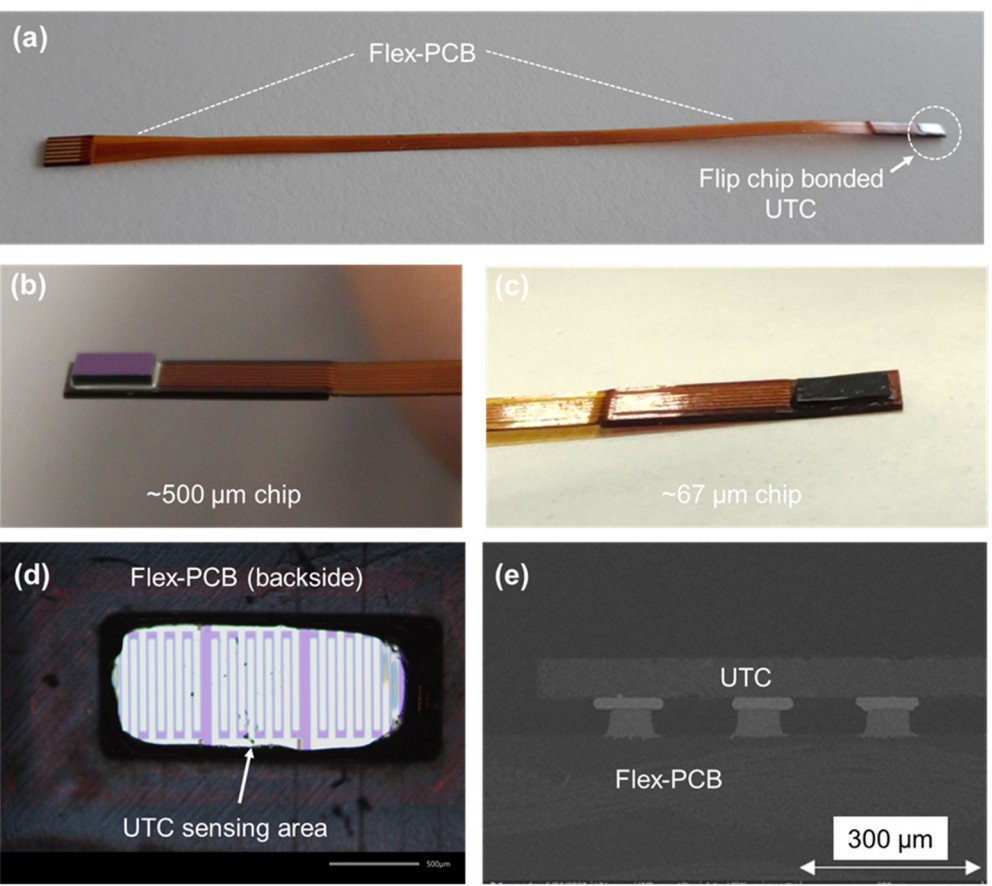

**Figure 5.** (**a**) Flip-chip integration of ultra-thinned chip (UTC) on flex with an exposed sensing area, (**b**) thick chip bonded on flex, (**c**) (UTC) bonded on flex, (**d**) exposed electrodes after bonding of UTC, and (**e**) SEM image of bonded UTC.

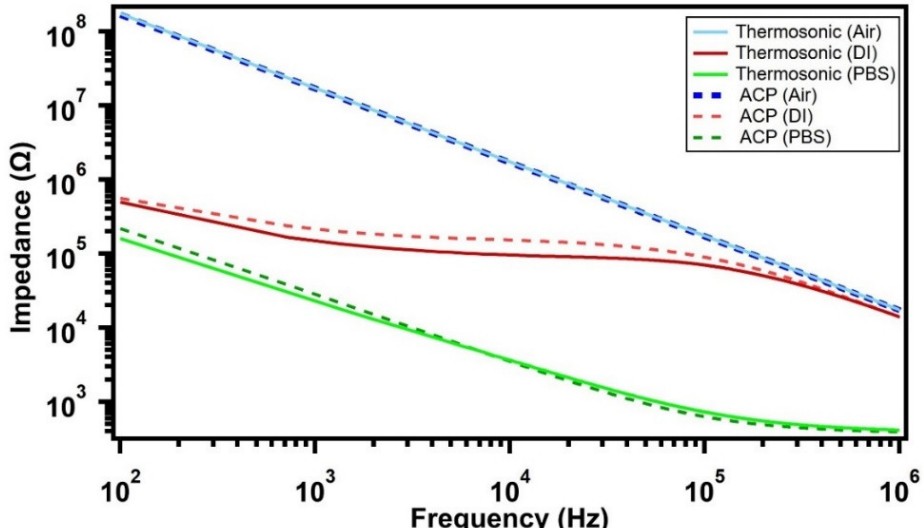

**Figure 6.** Impedance analysis of the ACP- and thermosonic-bonded samples in 3 mediums (air, Di water and PBS). The results revealed similar behavior up to 1 MHz.

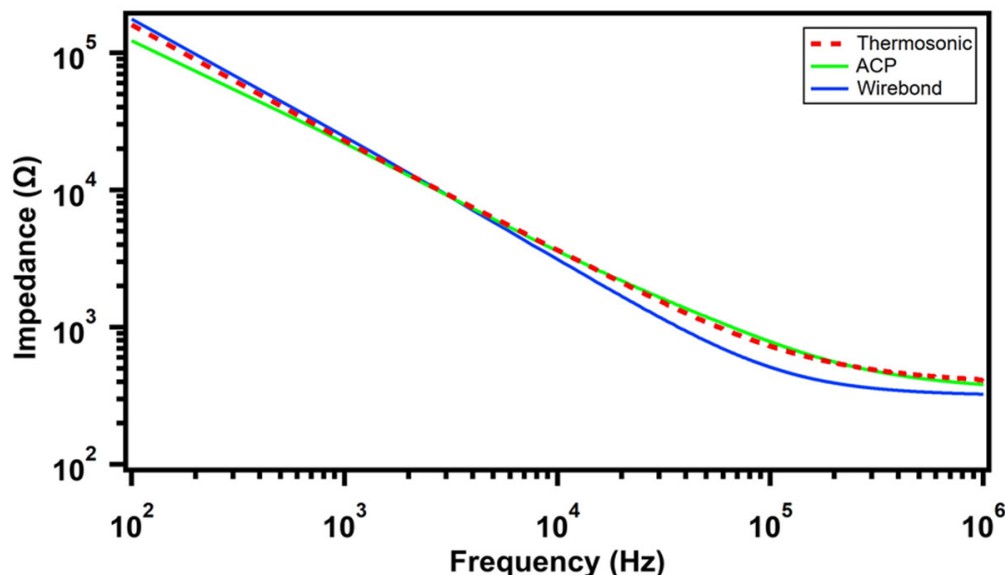

**Figure 7.** Wire-bonded (WB) sensor package in comparison to flip-chip thermosonc and ACP packages; initial reading implying similar behavior.

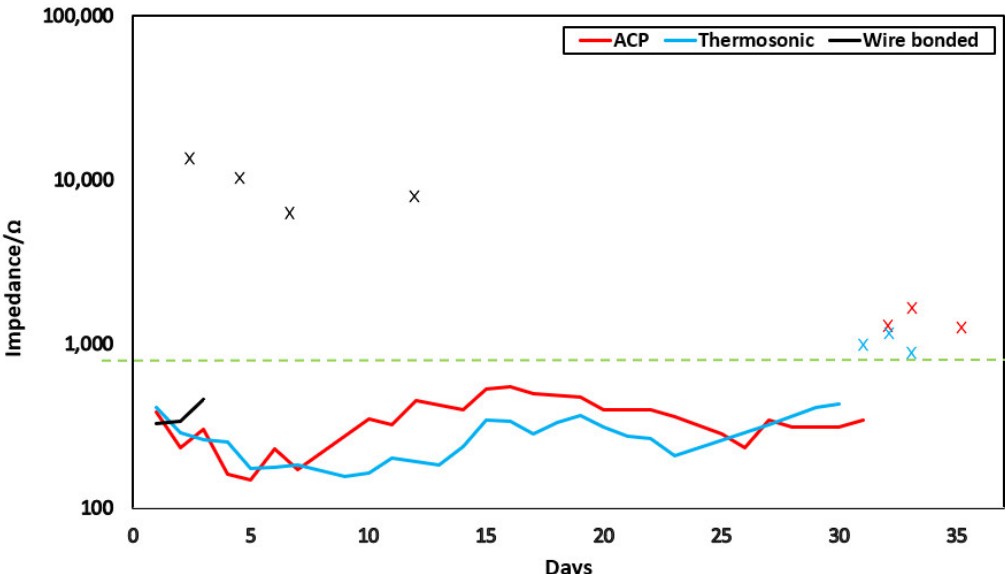

**Figure 8.** Long-term impedance variation of thinned IDE sensors far a span of days at 1 MHz, where (X) defines the failed samples.

In order to understand the root cause for the early failure of wire-bonded samples, the failed samples were analyzed (Figure 9). According to the optical and SEM cross-sectional analysis, it was postulated that the diffusion of PBS/water through the encapsulating epoxy (glob-top) and contamination of the sensitive aluminum bond pads was the reason. This moisture diffusion occurs due to water traversing the epoxy network through a network of nanopores that is inherent in the epoxy structure and through microcavities [29]. By comparing Figure 9a,b, one can clearly realize the degradation of the encapsulating glob-top and emergence of microcavities. For further investigation, cross-sectional analysis was carried out. A distinctive contaminated area was detected on the wall of the epoxy as shown in Figure 9c. EDX examination was performed to investigate the material composition of the contaminant. It was revealed that the contaminant comprises sodium (Na) and chlorine (Cl), which was in agreement with the hypothesis that the PBS diffused at the epoxy

interface, shown in Figure 9d,e. Moisture diffusion between the interface has previously been observed in literature by Haleh Ardebili [30] and can be an accelerator in the failure of the wire-bonded sample. The reason for the better performance of the flip-chip bonded samples was attributed to the better encapsulation of bond pads in flip-chip bonding. The lithographic and washing process to generate the cavity in the wire-bonded sample could have generated weak interfaces and caused susceptible spots in the package. Flip-chip bonded samples had a significantly small surface area in contact with the PBS compared to the epoxy in the wire-bonded sample [29]. Conclusively, it was proved that flip-chip bonded packages can prevail over conventional wire bonded samples and leverage the long-term reliability of the flexible sensor strips.

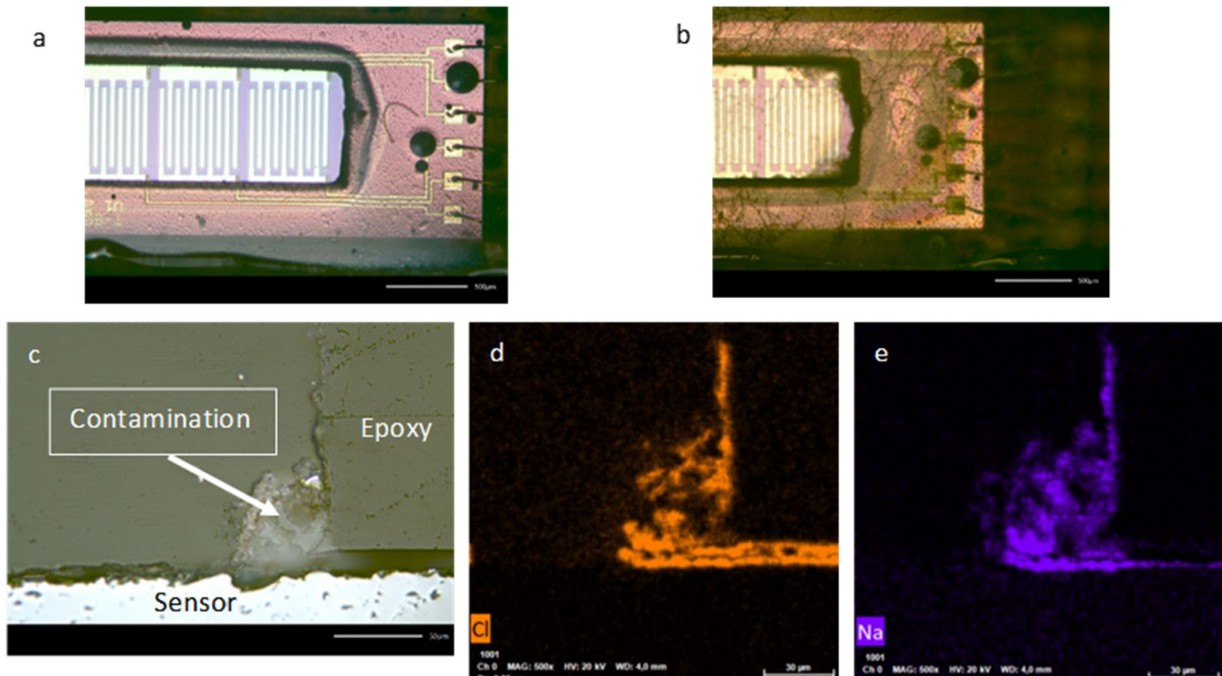

**Figure 9.** Failure analysis of the failed wire-bonded packages; (**a**) initial package, (**b**) failed package after 4th day of submergence in PBS solution, (**c**) cross-sectional optical image of the failed package and the degraded area, and (**d**,**e**) EDX images of the contaminated area in (**c**).

### 3.4. CMOS Chips

Figure 10a,b present the foundry fabricated unbumped (790 µm thick) and custom thinned unbumped (80 µm thick) CMOS chips, both ACP flip-chip bonded. Since CMOS chips are more sensitive to the induced stresses upon thinning, and bonding and pitch size were smaller than sensor chips, the functionality of the bonded thinned chips was a good indicator of the successful process flow. For the functionality test, a clock signal of 10 kHz and 3 V was applied to a digital buffer that would reproduce the input at the output. The results are shown in Figure 10c,d for both thick and thinned chips. The input signal is shown in green, while the blue (thinned) and red (original thickness) signals are the output. No functional problem was observed in any of the samples, nor did we notice any significant change in the power consumption. Hence, we can conclude that the process flow of thinning and integrating of CMOS MPW chips was successful.

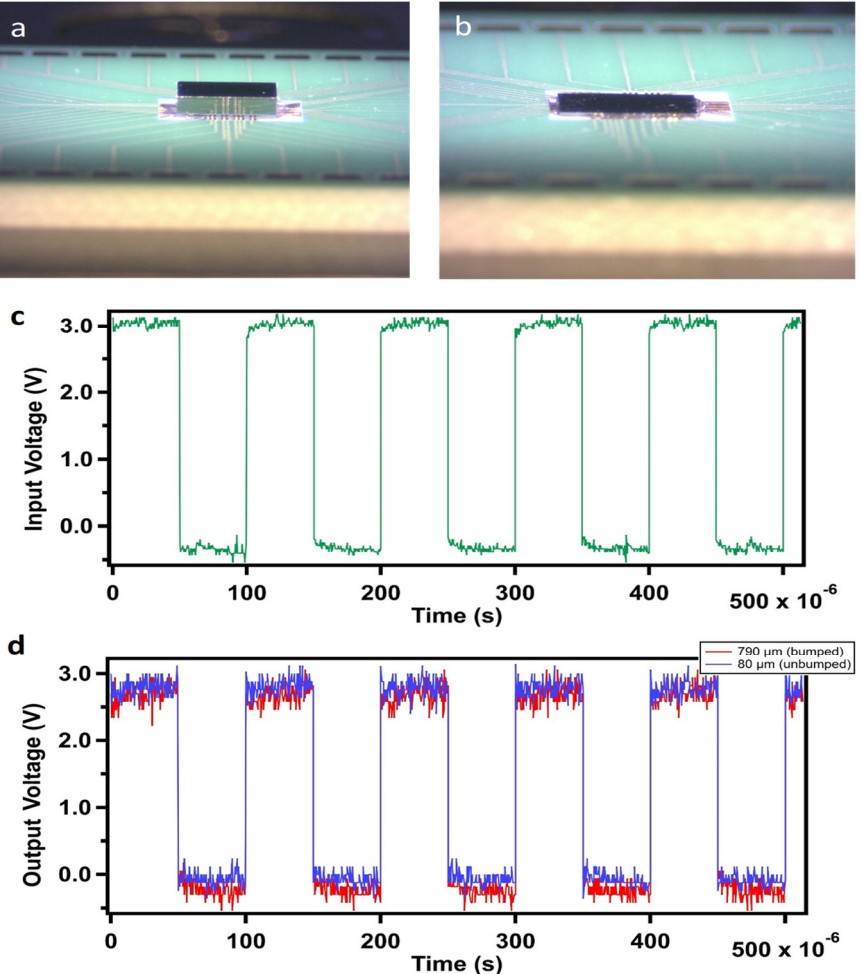

**Figure 10.** Die-level thinning of CMOS chips; a comparison between flip-chip bonded CMOS chips from MPW: (**a**) the original chips (with a thickness of 790 μm) and (**b**) die-level thinned chip (with a thickness of 80 μm ± 2 μm), (**c**) the input signal (green) and, (**d**) output of the CMOS chips, (790 μm in Red) and (80 μm in blue).

## 4. Conclusions

This paper detailed a die-level thinning and integrating route for singulated MPW chips using both silicon sensors and CMOS devices. The findings of this investigation may be summarized as follows. Die-level thinning of the chips from MPWs down to 80 μm was successfully performed by populating the dies on a silicon carrier wafer with a UV-curable adhesive film followed by back-grinding. The achieved total thickness variation was 2 μm. The thinned chips were integrated on a flexible substrate via flip-chip bonding by using low bonding forces. Based on the electrochemical impedance analysis of the sensor chips in PBS solution, it was revealed that both thermosonic and ACP flip-chip bonding rendered superior long-term performance compared to the conventional wire-bonded package. In the experimental evaluation, most wire-bonded packages failed due to the failure of the packages after 8 days on average, whereas for flip-chip bonding, the mean time to failure was observed around 30 days. It was concluded that the ACP flip-chip process is the most suitable process for integrating thin chips on flex with the least complexity and process steps. The ACP process does not require an additional ultrasonic source, and does not necessarily need bumping of the chip pads. Furthermore, similar steps were performed on a CMOS die where the thinning process and the integration process were also successfully performed and the achieved characteristics were compared with the original, non-thinned

dies. Hence, we validate a method to integrate singulated ultra-thin silicon/CMOS dies from MPW wafers on flexible substrates and in a harsh environment.

**Author Contributions:** Conceptualization: M.H.M. and A.R.; investigation: M.H.M., A.T. and A.R.; methodology: M.H.M., A.T., S.M. and A.R.; project administration: A.B. and A.R.; writing—original draft: M.H.M. and A.R.; writing—review and editing: M.H.M., A.T., H.Z., A.B, S.M. and A.R. All authors have read and agreed to the published version of the manuscript.

**Funding:** This project has received funding from the European Union's Horizon 2020 research and innovation program under the Marie Skłodowska-Curie grant agreement No.: H2020-MSCA-ITN-2018-813680.

**Institutional Review Board Statement:** Not applicable.

**Informed Consent Statement:** Not applicable.

**Data Availability Statement:** Data are available upon request from authors.

**Conflicts of Interest:** The authors declare no conflict of interest.

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
