# Peer review of "Die-Level Thinning for Flip-Chip Integration on Flexible Substrates"

_electronics, doi:10.3390/electronics11060849_

Round 1
Reviewer 1 Report
The core value of this work is its comparative study on three different bonding/integration methods. This touches down a fundamental technical challenge in flexible electronics and the results are of great interest to readers. I suggest the authors to tweak several places before publication.
- Reconsider its title. The current one seems focus on "thinning" rather than "flipchip integration", whereas the manuscript highlights the later.
- Introduction and Figure 1: more references/background knowledge is helpful to understand why "thinner" chips is favored on a flexible substrate: is it a result of the increasing flexibility of the chip itself, or because of a reduced mechanical mismatch at the chip/substrate interface upon bending?
- Cavity. More dimensional information is required, particular interest is the depth.
- Impedance results. None of integration methods ensures a same exposure area of the IDE electrode. This is likely to be one cause for impedance variations, and should be pointed out in the discussion.
- Failure bonding. In figure 9 c, providing a similar cross-sectional SEM of the Sensor-Epoxy interface before the immersion study can be straightforward to assess contamination issues.
Reviewer 2 Report
The manuscript reports the die-level thinning for flip-chip integration on flexible substrates. The results are interesting. However, The chips are not easy to handle. Some of comments have to address as following:
- I-V characteristics are required to discuss the leakage current after thinning treatment.
- Pictures of IDE and surface of die in magnification before and after thinning treatment are required. Any peeling, damage, and scratch?
- How about yield for die-level thinning?
- For flexible substrate, how about the electrical characteristics in bending situation?
Therefore, I recommend it as major revision to publish.
Author Response
Pleases see the attachment

Round 2
Reviewer 2 Report
The manuscript has revised well. Therefore, in my opinion, the article be accepted to publish.